# Heat Stress Responses and Thermotolerance in Maize

**DOI:** 10.3390/ijms22020948

**Published:** 2021-01-19

**Authors:** Zhaoxia Li, Stephen H. Howell

**Affiliations:** Plant Sciences Institute, Iowa State University, Ames, IA 50011, USA; zhaoxial@iastate.edu

**Keywords:** heat stress, transcriptional regulation, post-transcriptional regulation, maize

## Abstract

High temperatures causing heat stress disturb cellular homeostasis and impede growth and development in plants. Extensive agricultural losses are attributed to heat stress, often in combination with other stresses. Plants have evolved a variety of responses to heat stress to minimize damage and to protect themselves from further stress. A narrow temperature window separates growth from heat stress, and the range of temperatures conferring optimal growth often overlap with those producing heat stress. Heat stress induces a cytoplasmic heat stress response (HSR) in which heat shock transcription factors (HSFs) activate a constellation of genes encoding heat shock proteins (HSPs). Heat stress also induces the endoplasmic reticulum (ER)-localized unfolded protein response (UPR), which activates transcription factors that upregulate a different family of stress response genes. Heat stress also activates hormone responses and alternative RNA splicing, all of which may contribute to thermotolerance. Heat stress is often studied by subjecting plants to step increases in temperatures; however, more recent studies have demonstrated that heat shock responses occur under simulated field conditions in which temperatures are slowly ramped up to more moderate temperatures. Heat stress responses, assessed at a molecular level, could be used as traits for plant breeders to select for thermotolerance.

## 1. Introduction

As sessile organisms, plants are exposed to a variety of environmental conditions, some of which cause stress. One of the major stresses in crop plants is heat stress, which is usually accompanied by other stresses brought about by conditions such as drought or salinity [1]. Given the prediction for increases in average global temperature, plants will be faced with greater frequencies of high-temperature events (ICPP report [2]). Therefore, plant scientists and breeders are challenged to understand how plants, especially crop plants, can better tolerate heat stress. The responses of plants to heat stress have been extensively studied, but the means to confer tolerance is less well-understood. In recent years, comparative transcriptomics and metabolomics have been used to study heat stress responses in different maize lines. For example, in transcriptome studies of four heat-tolerant and four heat-susceptible inbred lines, 607 heat-responsive genes, as well as 39 heat-tolerance genes, were identified [3]. In another transcriptome study of heat tolerance in two sweet corn lines, the gene ontology (GO) terms of the biosynthesis of secondary metabolites, the upregulation of photosynthesis, and the downregulation of ribosome function correlated with improved heat resistance [4]. In order to identify metabolic markers of heat stress responses, metabolite profiles of maize leaves have been analyzed under adverse environmental conditions, including heat stress [5,6,7]. Similar to what has been found in Arabidopsis, altered energy pathways, increased production of branched-chain amino acids, raffinose family oligosaccharides, lipolysis products, and tocopherols have been found in response to heat stress [5,6,7]. Under severe stress conditions, the production has been observed of some metabolites that serve as osmolytes, antioxidants, and growth precursors and lipid metabolites that help protect membranes against heat stress. Many of the responses to stress conditions involve the induction of stress-responsive genes, while others involve post-transcriptional processes. In this review, the regulation of these responses, as they are known in maize, will be described.

## 2. Heat Shock and Heat Stress in Lab and Field Conditions

The effect of high temperature on the yield performance and stability of maize in the field has been the subject of a number of studies [8]. Lobell et al. [9] reported that, for each degree day spent above 30 °C, the final yield of African maize declined 1%. Likewise, Badu-Apraku et al. observed that corn yields declined 42% in grain weight per plant with increases in day/night temperatures from 25/15 to 35/15 °C from 18 d post-silking to maturity [10]. The impact of temperature depends not only on the severity of the stress but on its timing in the life cycle of the plant. Heat waves during reproductive development can have a major impact on production. The effects of high temperature during silking, pollination, and grain filling have profound effects on maize pollen shedding and viability [11,12] and, at later stages, on grain filling [10,13]. The optimal temperature range for maize growth is 25–33 °C, day/17–23 °C, night, which are mid-summer temperatures in the U.S. corn belt. Furthermore, the optimum temperature for maximum maize grain yield is around 25 °C (https://www.extension.purdue.edu/extmedia/nch/nch-40.html). Surprisingly, the onset of heat stress during vegetative growth appears to lie within the range of optimum growth conditions (Figure 1a). In one case where this has been studied (in an inbred line W22 grown under controlled conditions), the onset of heat stress was indicated by the activation of a heat shock protein gene, HSP26 [14]. Nonetheless, the overlap between growth and heat stress suggests that there is a tradeoff between the two, and at temperatures below some optimums, growth dominates, and above that, temperature heat stress outweighs growth. Various warming studies have been conducted under field conditions, such as the studies at the University of Illinois using infrared heaters deployed in the field. Siebers et al. [15] found that raising the canopy temperature by 6 °C in heat waves for three consecutive days during the V7 stage of growth had no significant effect on the reproductive biomass, but the reproductive biomass was reduced when heat waves were applied during silking. However, heat waves at younger vegetative stages had detrimental effects on the maize growth and development. Some of the highest temperatures relative to the average in the U.S. corn belt occurred in May 2012 [16], during which time, the crop conditions dropped significantly during early vegetative growth (http://www.kingcorn.org/news/articles.12/HotDryMidSeason-0625.html). 

Crop plants, such as maize, have not been well-studied under controlled environmental conditions largely because of the high light requirement and the difficulty in simulating field conditions in greenhouses or growth chambers (see, for example, Pooter et al., 2016 [17]). However, the advantage of controlled conditions is to minimize the environmental variables and to focus on one or a few parameters that may affect plant responses. The object of controlled environment studies is to observe a physiological or molecular response to a given environmental condition, not to predict performances in the field affecting complex traits such as yield. In the lab or growth chamber conditions, the effects of heat waves are often simulated by heat shock treatment. These conditions are not natural, because they typically involve a step increase of the temperatures that are much higher than normal growth conditions, usually above the growth threshold for that species [18]. However, for a better simulation of field conditions, plants in growth chambers can be exposed to moderately elevated temperatures in diurnal cycles for several days, during which time, the temperatures can be slowly ramped up and down as they occur in field conditions (Figure 1b). The developmental stage at which studies are conducted is also an important parameter. For maize, rice, and wheat, high temperatures at the flowering and grain-filling stages limit production [13,19,20]. Elevated temperatures at those stages affect pollen viability, leading to slower pollen growth, poor fertilization, abnormal ovary development, kernel abortion, and/or limited grain production.

## 3. Heat Stress Response (HSR), A Cytoplasmic Heat Stress Response

Heat stress triggers protective mechanisms collectively called heat stress responses (HSRs). HSRs are homeostatic mechanisms to mitigate the damage from heat stress and to protect plants from further stress. Heat shock factors (HSFs) are transcription factors that play a central role in HSRs by activating the expression of constellations of heat shock proteins (HSPs) (Figure 2 and Table 1). HSF activity is regulated at multiple levels, including the transcriptional, post-transcriptional, translational, and post-translation levels (reviewed by [21,22]). Under nonstress conditions, class A1 HSFs are sequestered in the cytoplasm by their association with HSP90/70 and their cochaperones. In response to heat stress, misfolded proteins recruit chaperones away from HSFs [23], and the liberated HSFs undergo trimerization and are imported into the nucleus [24]. HSF trimerization promotes the ability of HSFs to bind to heat shock response elements (HSEs) on HSP gene promoters and activate their transcription [25,26]. Plant HSF genes belong to moderate-sized families of HSF genes, with 21 members in Arabidopsis [27], 25 members in rice [28], and 31 members in maize [29,30]. Plant HSFs have conserved modular structures, in which their N-terminal domains bear helix-loop-helix DNA-binding motifs and oligomerization domains with bipartite heptad patterns of hydrophobic amino acid residues (HR-A/B region) [27,28,29,30,31]. These two domains are connected by a flexible linker, and based on the length of the linker, plant HSFs are classified into three classes: HSFA, B, and C [27,32]. The C-terminal domains of plant class A HSFs are activation domains characterized by short peptide motifs (AHA motifs), which are crucial for transcriptional activation [27,32]. The class A HSFs have AHA motifs and operate as transcriptional activators, while the class B and C HSFs do not have AHA motifs and do not have activator functions on their own [33].

Unlike mammalian cells, most plant HSFs are themselves regulated by heat shock; some are upregulated, while others are downregulated. In Arabidopsis, HSFAs constitute a transcriptional cascade in which HSFs activate each other in response to environmental stress (Figure 3a and Table 1). Nishizawa-Yokoi et al. [34] discovered the cascade by creating chimeric repressors of all the HSFA proteins in Arabidopsis. The regulation of HSF2A was of central interest, because gene expression profiling revealed that HSFA2 was the most highly upregulated HSFA gene in response to heat stress [35]. Furthermore, *HSFA2* overexpression gives rise to transgenic plants with a greater tolerance to environmental stress, and *HSFA2* knockouts reduce the thermotolerance [35,36,37,38]. Nishizawa-Yokoi et al. [34] found that the expression of *HSFA1d* and *HSFA1e* was linked to the upregulation of *HSFA2* in response to environmental stress, but that the expression of related *HSFA1a* and *HSFA1b* was not. *HSFA2* itself was also involved in the upregulation of other HSF genes, as well as HSP genes. Hence, in Arabidopsis, the transcription of HSFs plays an important role in the HSR and in conferring thermotolerance.

Nine HSFs were upregulated at elevated temperatures in the leaves of maize inbred line W22 plants (at stage V4 and V5 plants) grown under controlled conditions in the Enviratron, including one type A HSF, *Hsftf13* (Zm00001d027757, named as *ZmHsf01* in [29] and [30,82]), which was most highly induced [14] (summarized in Figure 3b). Maize *Hsftf13* is most closely related to *AtHSF6b* in Arabidopsis, which plays a pivotal role in responses to abscisic acid (ABA) and thermotolerance [40]. *HSFA6b* responds to ABA signaling via the AREB/ABF-ABRE regulon and activates the transcriptional activity of *HSP18.1-CI*, *DREB2A* (dehydration-responsive element-binding protein 2A) and *APX* (cytosolic ascorbate peroxidase), two promoters [40]. In phylogenetic terms, *Hsftf13* (Zm00001d027757) belongs to subgroup ATHSFA6,7, HSFs that have a single C-terminal activation (AHA) domain (Figure 4). The C-terminal activation domain of HSFA2 in tomatoes contains two independent activator motifs, much like its counterpart in Arabidopsis (AtHSFA2). The knockout of either domain in tomatoes resulted in a 20% to 50% reduction in activation potential [39]. Nonetheless, HSFTF13 has only one AHA domain, like the Arabidopsis subgroup members AtHSFA6a, AtHSFA6b, AtHSFA7a, and AtHSFA7b. *Hsftf13* in maize may function much like HAFA2 in Arabidopsis, based on their high induction by heat and activation motif content. Besides *Hsftf13*, four other HSFs were upregulated in both step heat shock and ramped heat stress conditions, including another A2 subgroup HSF (*ZmHsf04*, Zm00001d032923), an A6 subgroup HSF (*ZmHsf23*, Zm00001d046204), and two B2 subgroup HSFs (*ZmHsf11*, Zm00001d046204 and ZmHsf25, Zm00001d026094) (Figure 3b). Four HSFs, including two B1 and two C2 HSFs, were upregulated in response to ramped heat stress conditions but, surprisingly, not to step heat shock conditions. Compared with the group A HSFs, which are characterized by having one or two AHA motifs, most of the group B and C HSFs lack AHA motifs [27,33], the core activator domains of class A HSFs. It is not clear why these HSFs respond specifically to simulated field heat stress conditions but not to step heat shock treatment. Perhaps it is because these HSFs are involved in slow-resolving dimerization or complex formation. In Arabidopsis, three B-type HSFs, *HSFB1*, *HSFB2a*, and *HSFB2b,* are heat-inducible, and these HSFBs have repressor activity [41,42]. Interestingly, the induction of *HSFA2*, *HSFA7a*, *HSFB2b*, *HSP15.7CI*, and endogenous *HSFB1* expression was suppressed or reduced in Pro35S:HSFB1 seedlings but not in Pro35S: mutHSFB1 or wild-type seedlings under the moderate heat condition (28 °C). The different expression patterns of the group A, B, and C HSFs suggest that plants have mechanisms to cope with the different levels of heat stress, and the HSFBs could be factors that shape the different transcriptomes in response to natural/moderate heat stress and heat shock treatment.

HSFs activate the expression of HSPs in response to heat stress but, also, in response to other stresses [83,84,85]. HSPs are grouped in plants into five classes according to size: HSP100, HSP90, HSP70, HSP60, and small heat shock proteins (sHSPs) [86,87,88]. Generally, HSPs are thought to function as chaperones to alleviate protein misfolding and aggregation problems. In maize, as in Arabidopsis, heat shock activates the expression of both high and low molecular weight HSPs [3,43,44,45,46]. The proteins in maize plants produced in response to heat shock were identified through a proteome analysis and included such proteins as the ATPase beta subunit, HSP26, HSP16.9, and unknown HSP/chaperonin [47,48]. It was actually surprising to find that *Hsp26,* an early heat stress marker gene, and *Hsp17.6* and *Hsp18.2* were upregulated under the simulated field conditions when the temperatures were slowly ramped up [14]. One might think that plants would acclimate to the high temperature by slowly ramping up the higher temperature conditions [89]. This would suggest that, on a hot summer day, the small HSP genes are deployed even when the temperature rises slowly in the morning to a peak in the afternoon. The small HSPs are often the first line of defense in response to other stresses, including drought and salt stress, earning the title as the “paramedics of the cell” [90]. HSP70, HSP90, and HSP101, together with a mitochondrial HSP60, were also induced by heat in diurnal temperature cycles [14].

## 4. Plant UPR in Heat Stress Response

Besides the HSR, another response called the unfolded protein response (UPR) in the endoplasmic reticulum (ER) is also activated by heat and helps to mitigate the damage caused by heat stress and to protect plants against further stress conditions (Figure 2 and Table 1). These responses are partly transcriptional and partly post-transcriptional. UPR is activated by misfolded proteins that accumulate in the ER under adverse environmental conditions. When the demand for protein folding exceeds the capacity of the system, UPR is initiated. Functional crosstalk between UPR and plant hormone signaling was also observed in plants to cope with the adverse environment [91,92]. The UPR is an evolutionarily conserved system that protects plants from stress. The UPR and HSR occur in different cellular compartments, but both responses are elicited by misfolded proteins that accumulate in the ER and cytoplasm, respectively (Figure 2). The UPR upregulates the expression of a canonical set of genes to resolve the protein folding problem by enhancing ER protein import, folding, quality control, and export. The UPR genes in plants are upregulated by the activation of stress-transducing transcription factors [93] through two branches of the UPR signaling pathway. One branch involves the dual protein kinase and ribonuclease, IRE1, which splices basic leucine zipper (bZIP) into 60 mRNA when activated. The other branch is mediated by two ER membrane-anchored transcription factors, bZIP17 and bZIP28 [93]. Both branches respond and contribute to the protection from heat stress. For example, bZIP60s activates the expression of a type A HSF, *Hsftf13*, which, in turn, upregulates the expression of a constellation of HSP genes [14]. bZIP28, which encodes an ER membrane-associated bZIP transcription factor, contributes to the upregulation of heat-responsive genes and to heat tolerance [49,50]. bZIP28 binds directly to the promoters of heat-responsive genes in a tissue-specific manner [50]. IRE1, a major component of the UPR, plays an important role in protecting pollen development from elevated temperatures [51,52].

In Arabidopsis, heat shock activates both bZIP28 [49] and bZIP60 [51] on the two different branches of the UPR signaling pathway, which leads to the upregulation of canonical UPR genes and noncanonical genes, including HSR genes [49,50]. bZIP60 appears to have been under selection during the maize domestication and acclimatization from tropical to temperate conditions, because higher expression levels of bZIP60 were observed in tropical maize lines compared to temperate lines. These differences are correlated with the presence of transposable elements in the promoter of bZIP60 [53].

## 5. Other Transcriptional Regulation Networks Involved in Plant Heat Stress Response

Thermotolerance can be acquired in plants through the action of other signaling/response pathways (Table 1), such as by the pretreatment with ABA (abscisic acid), SA (salicylic acid), ACC (1-aminocyclopropane-1-carboxylic acid, an ethylene precursor), or H_2_O_2_. Mutants in the biosynthetic or response pathways for these compounds are defective in basal thermotolerance [94]. ABA is believed to play roles in heat acclimation-induced heat tolerance or acquired heat tolerance. ABA-deficient and -insensitive mutants are sensitive to heat, whereas AREB/ABF-OE plants show enhanced thermotolerance [54,55,56,94]. ABA treatment and AREB1/ABF2 co-expression enhanced the *HSFA6b* transcription activity in a protoplast transcriptional activation assay. In this system, *HSFA6b* directly bound to the promoter of *DREB2A* and enhanced its expression [40]. The upregulation of *HSFA2c* and *HSPs* by ABA contributed to an improved heat tolerance in tall fescue and Arabidopsis through the direct binding of FaDREB2A or FaAREB3 to the ABRE cis-elements in the promoter of *FaHSFA2c* [58]. ABA treatment significantly increased the thermotolerance in maize seedlings by enabling the maintenance of higher of antioxidant enzyme activities and lower levels of lipid peroxidation [57]. Many of the transcripts that accumulated in plants in response to a combination of salt and heat stress involved ABA, and the mutants deficient in ABA metabolism (*aba1*) and signaling (*abi1*) were found to be more susceptible to these stresses [56]. *ERF1 (ETHYLENE RESPONSE FACTOR1*), an upstream component in both jasmonic acid (JA) and ethylene signaling, activated *HSFA3* and *HSP70* expression and enhanced the thermotolerance in Arabidopsis [59]. In another study, JA was thought to play a key role in the response of plants to a combination of heat stress and high light, because the levels of JA and its conjugate increased dramatically in response to the stress combination, and JA-deficient allene oxide synthase (*aos*) mutants were more sensitive to this same combination of stresses [60]. Altered systemic reactive oxygen species (ROS) signaling during the high light and heat stress combination in the *aos* mutant supported the role of JA in suppressing the initiation of the ROS wave in local leaves of plants treated with heat stress and high light [61].

Other transcription factors (TFs) and DNA methylation were also found to be involved in the heat stress responses (Table 1). DREB2A induces the expression of dehydration- and heat stress-inducible genes under stress conditions. The overexpression of DREB2A-CA (constitutively active form) significantly enhanced the heat stress tolerance, and DREB2A knockout mutants exhibited a significant decrease in heat stress tolerance [62]. NUCLEAR FACTOR Y, SUBUNIT C10 (NF-YC10/DNA POLYMERASE II SUBUNIT B3-1, DPB3-1) was identified as a DREB2A interactor. NF-YC10 formed a transcriptional complex with the NF-YA and NF-YB subunits, and the trimer enhanced the heat stress-inducible gene expression in cooperation with DREB2A [63]. Interestingly, NF-YC10 is also recognized as a moonlighting glyceraldehyde-3-phosphate dehydrogenase (GAPC)-binding transcription factor (see Figure 5). The overexpression of GAPC enhanced the heat tolerance of seedlings and the expression of heat-inducible genes, whereas a knockout of GAPCs had the opposite effect. Heat responses require GAPC and NF-YC10 interactions, and GAPC, which is normally located in the cytosol, enters the nucleus in response to heat. In the nucleus, GAPC binds to the transcription factor NF-YC10 to increase the expression of heat-inducible genes, rendering Arabidopsis tolerant to heat stress [64]. Two of the SQUAMOSA PROMOTER BINDING PROTEIN-LIKE (SPL) transcriptional factors in Arabidopsis, SPL1 and SPL12 act redundantly in conferring thermotolerance at the reproductive stage, and the acquisition of thermotolerance in the inflorescences requires PYL-mediated ABA signaling [65]. The overexpression of *OsMYB55* in maize improves the plant tolerance to high temperatures and water stress during vegetative growth by the activation of stress-responsive genes such as thaumatin, *Hsftf19*, two *WRKY* proteins, and five *MYB* proteins genes [95]. Recently, histone H3K4 methyltransferases SDG25 and ATX1 were found to be involved in the thermotolerance to moderately high temperatures and in the transcriptional memory of heat stress in Arabidopsis. SDG25 and ATX1 regulate the histone H3K4me3 level and prevent DNA methylation at loci associated with heat stress gene expression during stress recovery [66].

A major consequence of heat stress is the generation of excess reactive oxygen species (ROS), which leads to oxidative stress [96,97,98]. Heat-induced excessive ROS production occurs in different cell compartments, in chloroplasts, the mitochondria, plasma membranes, peroxisomes, apoplast, the ER, and cell walls. Similar to other environmental stresses, excessive ROS induces oxidative damage in lipids, proteins, and DNA. The toxic nature of ROS, the damage it causes, and the well-outfitted antioxidant machinery that defends against it, as well as the evidence that ROS also serves as a signaling molecule, have been extensively reviewed [96,97,98,99,100,101,102,103,104]. HSFs act as direct sensors of H_2_O_2_ in *Drosophila melanogaster* and mammalian cells, as shown by the induction of oligomerization and the acquisition of DNA binding of HSFs by H_2_O_2_ both in vivo and in vitro [105,106]. In plants, Davletova et al. [67], using dominant-negative constructs, demonstrated that HSF21 (named AtHSFA4a by Nover et al. [27], an A-class HSF) plays a key role in early H_2_O_2_ stress sensing in KO-*Apx1* plants (cytosolic ascorbate peroxidase). In Arabidopsis cell cultures, an increase in H_2_O_2_ production during the early phase of heat stress leads to the HSF-mediated induction of *HSP17.6* and *HSP18.2* [68]. In this system, the heat induction of the HSP gene expression was significantly reduced in the presence of the ROS scavengers/inhibitors ascorbate or diphenyleneiodonium chloride (DPI, an inhibitor of NADPH oxidase). In addition, NADPH oxidase mutants (*atrbohB* and D) in Arabidopsis reduced the thermotolerance [94], and the induction of HSR genes by light stress was much higher in knockout *Apx1* plants compared to that in wild-type plants [69]. These studies support the role of ROS in helping plants to cope with heat stress.

## 6. Enhancement of Alternative RNA Splicing As a Heat Stress Response

Heat also enhances alternative RNA splicing (AS) in maize. AS produces different RNA isoforms, thereby enhancing the complexity of the genic output and, possibly, the repertoire of responses to different environmental conditions [107,108,109]. In a study conducted under controlled conditions in the Enviratron, elevated daily temperatures enhanced the frequency of global AS events in maize [70] (Li, in review). Other studies showed that AS affects HSR in plants (Table 1). In Arabidopsis, heat stress-induced AS leads to the production of a full-length *HSFA2* transcript [71] by skipping a cryptic mini-exon that lies within the *HSFA2* intron and that has a premature stop codon (PTC). An additional splice variant *HSFA2-III* is produced at higher temperatures (42–45 °C) because of the use of a cryptic 5′ splice site in the intron [72]. *HSFA2-III* encodes a truncated protein, S-HSFA2, which can bind to the *HSFA2* promoter, upregulating HSFA2 expression and constituting a positive autoregulatory loop. In maize, Zhang et al. [30] observed slight increases in the alternatively spliced forms of *ZmHsf04* and *ZmHsf17* following heat shock (42 °C). In other maize studies conducted in the Enviratron, elevated temperatures produced a greater abundance of *Hsftf9* RNA isoforms with longer 3′-UTRs. As discussed above, the temperature elevation in the Enviratron experiments were conducted by ramping up the temperature slowly to 35 and 37 °C, simulating the conditions on a summer morning through midday on a hot summer day in the Corn belt.

The other genes most frequently subjected to AS at elevated temperature are the genes encoding splicing factors themselves, especially the genes encoding splicing regulators, serine/arginine-rich (SR) proteins. AS of pre-mRNAs of Arabidopsis SR genes produced different isoforms likely to encode proteins with altered functions in pre-mRNA splicing. Heat stress in Arabidopsis (38 °C for 6 h) dramatically altered the pattern of AS of pre-mRNAs of several SR genes, more so than the effects of stressors/hormones, such as ABA, IAA, 6-BA, NaCl, and cold stress [73]. In maize, one of the major genes involved in AS, SR45a, underwent AS changes throughout the day, leading to the production of different SR45a RNA isoforms. In general, SR45a RNA isoforms capable of encoding proteins with greater splicing efficiencies were produced later in the day as temperatures rose. SR proteins are key players in the regulation of AS [70] (Li, in review). The AS of SR genes increased the complexity of the SR gene family transcriptome, although a number of these splice variants were nonproductive, encoding proteins lacking functional domains.

Another well-known gene subject to temperature-dependent AS is *FLOWERING LOCUS M* (*FLM*), a component of the thermo-sensory flowering pathway in Arabidopsis [74]. FLM acts as a “thermometer”, measuring moderate changes in the ambient temperature. Two splice variants of FLM, namely FLM-β and FLM-δ, in Arabidopsis result from the temperature-dependent AS of FLM pre-mRNA via a mutually exclusive event (*FLM-β* mRNAs after splicing contains exon 1 (E1), E2, E4, and E5, while *FLM-δ* contains E1, E3, E4, and E5) involving the second and third exon [75] (see Figure 5b). The abundance of *FLM-β* relative to *FLM-δ* changes in response to temperature [76,77]. The ratio of the two splicing isoforms, *FLM-β/FLM-δ*, fine-tunes the flowering time and is regulated by an ambient temperature [76,77]. FLM creates a large number of splice variants at high temperatures (27 °C) besides *FLM-β*, which are targeted for destruction by nonsense-mediated mRNA decay (NMD) [74,78].

MicroRNA levels are also affected by the temperature. Squamosa-binding protein-like (SPL) transcription factors are targeted by microRNA156 (miR156), and heat stress induces the expression of miR156 in Arabidopsis [79,110]. miR156s contribute to heat shock memory and influence the expression of HS memory-related genes through their effect on SPL genes [79]. That is why ARGONAUTE1, an RNA slicer that selectively recruits microRNAs and small interfering RNAs (siRNAs), is required to maintain the acquired thermotolerance. Besides miR156, heat stress affects other miRNAs and their target genes involved in protein folding, antioxidant response, photosynthesis protection, reproductive tissue protection, and flowering time regulation [80,111]. In maize, 61 known miRNAs belonging to 26 miRNA families and 42 novel miRNAs showed significant differential expression in the response to heat stress, with the majority being downregulated [81,112]. The reduction in miRNA levels leads to the upregulation of their targets, such as in the case of miR172-AP2.

## 7. Conclusions, Prospects and Maize Breeding

Although our attention was focused on individual genes in response to heat stress, we do not have a holistic view of how heat tolerance in plants is brought about. Nonetheless, we are learning more about how plants sense temperature and turn it into a signal to control certain transcription, post-transcription, or translational events. It is of interest as to how two evolutionarily conserved cellular systems in the ER and cytoplasm communicate about heat stress with each other and elicit secondary signals involving calcium or ROS. We want to know how systemic signals are transmitted from leaves to “sink” organs such as the ear to regulate growth and development in response to adverse environments. Additionally, what are the differences in the way plants deal with heat stress or heat stress combined with drought, salinity, or other stresses?

Maize is a major crop worldwide, and its production and yield stability are greatly affected by drought and high-temperature stresses. Due to the continuous increase in average global temperature, plants will more frequently face conditions of heat stress. Improving heat tolerance is a high priority in maize breeding programs, and the work of researchers in the lab should go hand-in-hand with the effort of breeders in the field. Maize has an abundance of heat tolerance variations in the native germplasm, since maize originated from the tropics and spread worldwide [113,114,115,116]. The characterization of the heat-tolerant inbred lines and the estimation of the breeding potential for creating heat-tolerant hybrids from these inbred lines is a challenge for maize breeders. The heat stress responses in maize described in this review could be used as traits to help breeders predict and identify heat-tolerant lines.

## Figures and Tables

**Figure 1 ijms-22-00948-f001:**
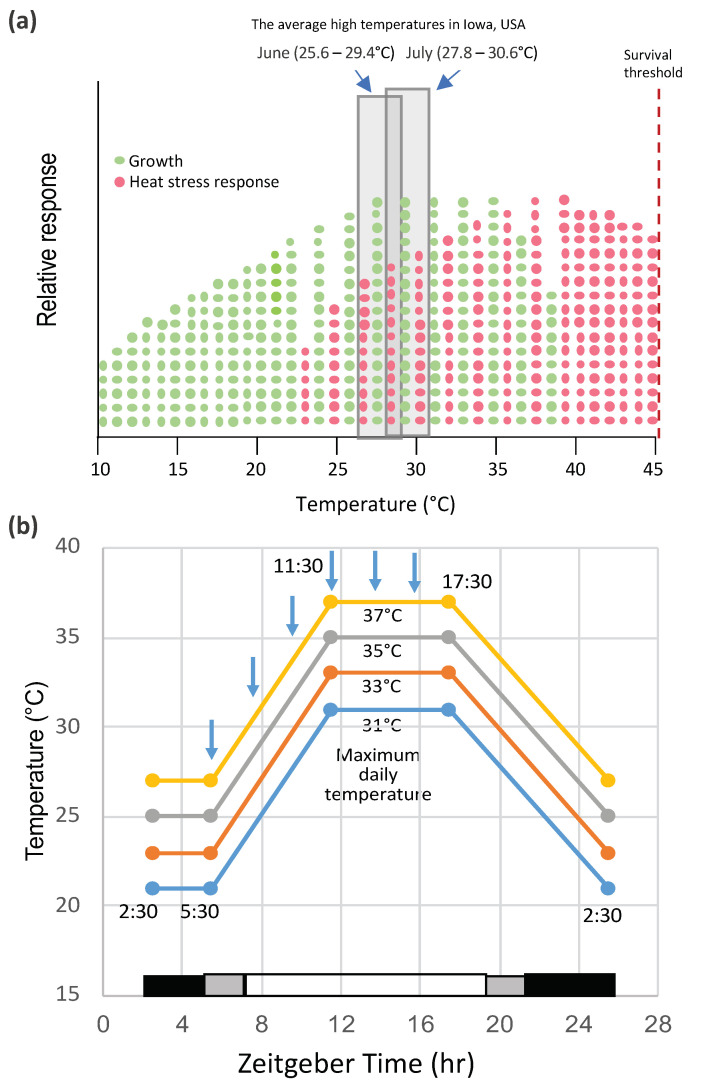
The relationship of temperature, growth, and heat stress in maize. (**a**) Optimal growth conditions overlap with heat stress. The optimal temperature range for maize growth is 25–33 °C, day/17–23 °C, night (https://www.extension.purdue.edu/extmedia/nch/nch-40.html), which are mid-summer temperatures in the US corn belt. The box indicates the average high temperatures (maximum daily temperatures) in June and July in Iowa, USA, a region representing an ideal climate for maize production. The trends shown here are from experiments conducted with the maize inbred line W22 grown under controlled environment conditions in the Enviratron at Iowa State University [14]. (**b**) An example of the temperature conditions used in the Enviratron to simulate diurnal temperature cycles under field conditions. Note the ramping up of the temperature to different maximum daily temperatures in the virtual afternoon and the ramping down in the virtual evening.

**Figure 2 ijms-22-00948-f002:**
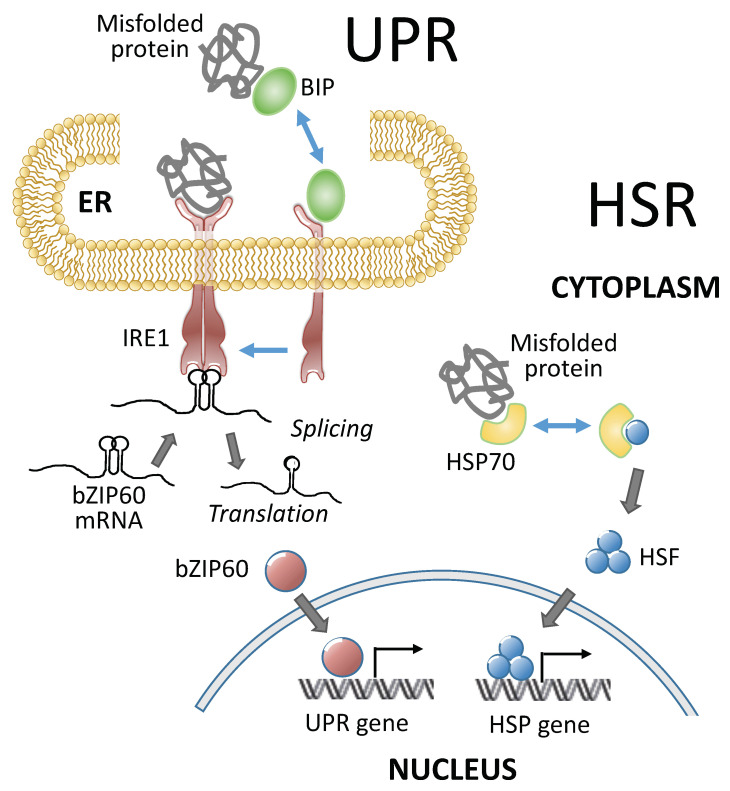
Two systems involved in heat stress responses in plants. Heat stress triggers protective mechanisms collectively called heat stress responses (HSRs). Both HSR in the cytoplasm and unfolded protein responses (UPR) in the endoplasmic reticulum (ER) mitigate the damage from heat stress and protect plants from further stress. The UPR and HSR occur in different cellular compartments, but both responses are elicited by misfolded proteins that accumulate in the ER and cytoplasm, respectively. The HSP and UPR genes in plants are upregulated by the activation of stress-transducing transcription factors, such as bZIP60 (basic leucine zipper 60) in the UPR and HSFs in the HSR. IRE1, a key factor in the UPR, is a dual protein kinase and ribonuclease involved in the splicing of bZIP60 mRNA. HSR, heat stress response; UPR, unfolded protein response; ER, endoplasmic reticulum; bZIP60, basic leucine zipper 60; HSP, heat shock protein; HSF, heat shock transcription factor; HSP70, heat shock protein 70; BIP, binding immunoglobulin protein; and IRE1, inositol-requiring enzyme 1.

**Figure 3 ijms-22-00948-f003:**
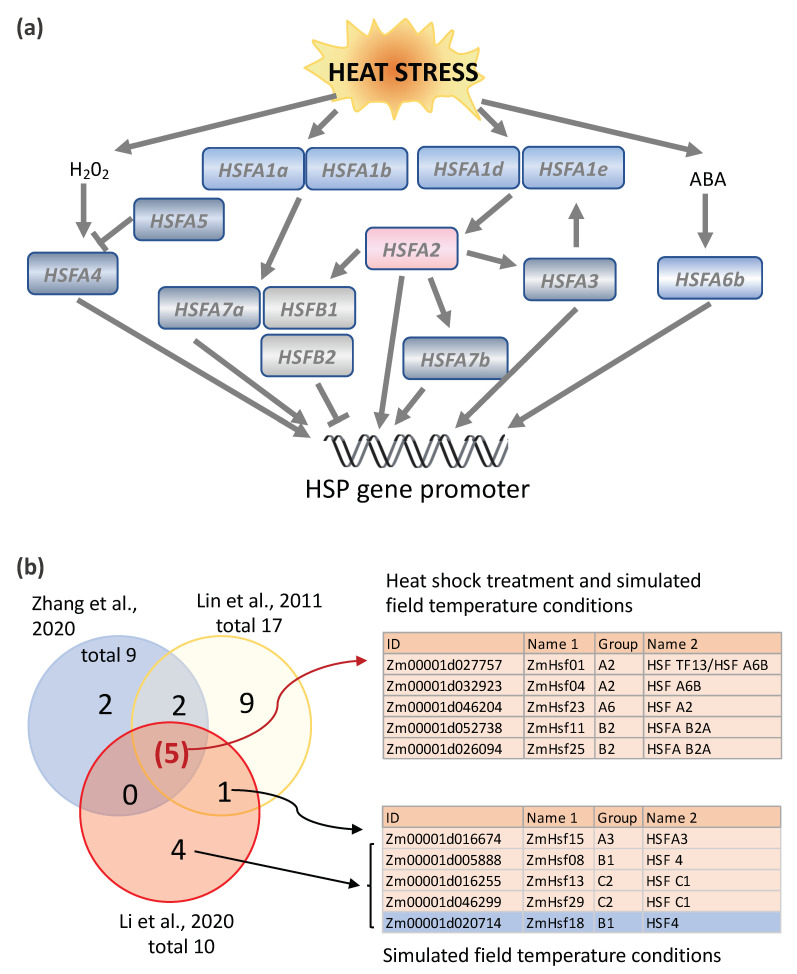
Heat stress induction of the HSF genes in Arabidopsis and maize. (**a**) Heat stress induction of a transcriptional cascade involving HSFA genes in Arabidopsis. Modified from Nishizawa-Yokoi et al., 2011 [34] and Huang et al., 2016 [40]. (**b**) Comparison of HSF induction in response to step heat shock treatment (Lin et al., 2011 [29] and Zhang et al., 2020 [30]) vs. ramped heat stress conditions (simulated field conditions in which temperatures are slowly ramped up to maximum daily temperature), Li et al., 2020 [14]). Five of the 31 maize HSFs (light orange color indicated) were identified as induced under both the step heat shock treatment and ramped heat stress conditions in maize, including two A2 subgroup HSFs, two B2 HSFs, and one A6 subgroup HSF. One B1 subgroup HSF and two C1 subgroup HSFs (light orange) were specifically upregulated in response to ramped heat stress conditions, while one B1 HSF (blue) was downregulated under the same conditions. HSFA, class A heat shock transcription factor; HSFB, class B heat shock transcription factor; ABA, abscisic acid; HSP, heat shock protein; and HSF TF, heat shock transcription factor.

**Figure 4 ijms-22-00948-f004:**
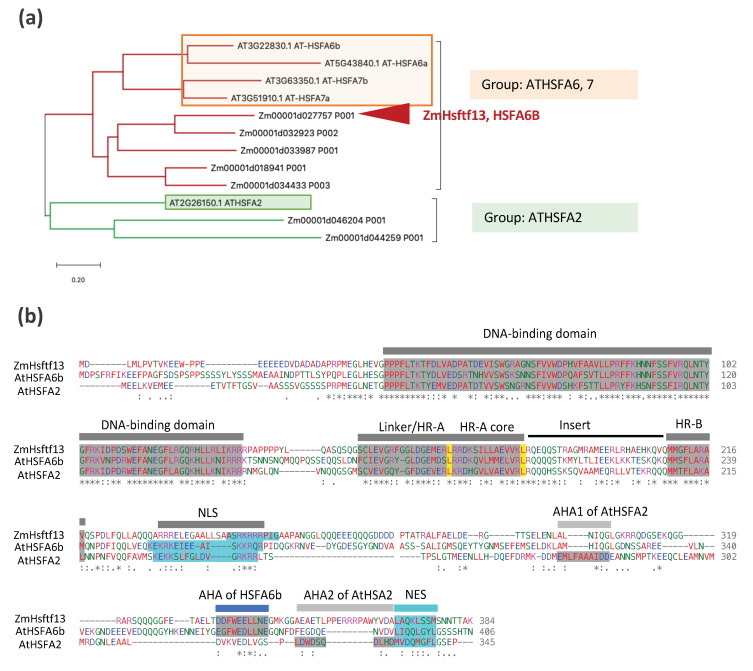
Phylogenetic relationship and domain analysis of Hsftf13. (**a**) Amino acid sequence alignment of the Hsftf13 subgroup genes in Arabidopsis and maize. In this clade, there is two subfamily HSF genes in Arabidopsis (HSFA2 and HSFA6,7). Two maize HSFs in subgroup HSFA2 and five HSFs in the HSFA6,7 subgroup, including Hsftf13. The evolutionary analysis was conducted in MEGA X (https://www.megasoftware.net) by using the maximum likelihood method and JTT (Jones-Taylor-Thornton) matrix-based model. The tree with the highest log likelihood is shown. (**b**) Amino acid sequence alignment and major domains in those HSFs. AHA motifs, activation domain, and characterized by short peptide motifs. The oligomerization domain HR-A/B is characterized by the heptad pattern of hydrophobic residues. NLS represents nuclear localization signal, and NES indicates nuclear export signal.

**Figure 5 ijms-22-00948-f005:**
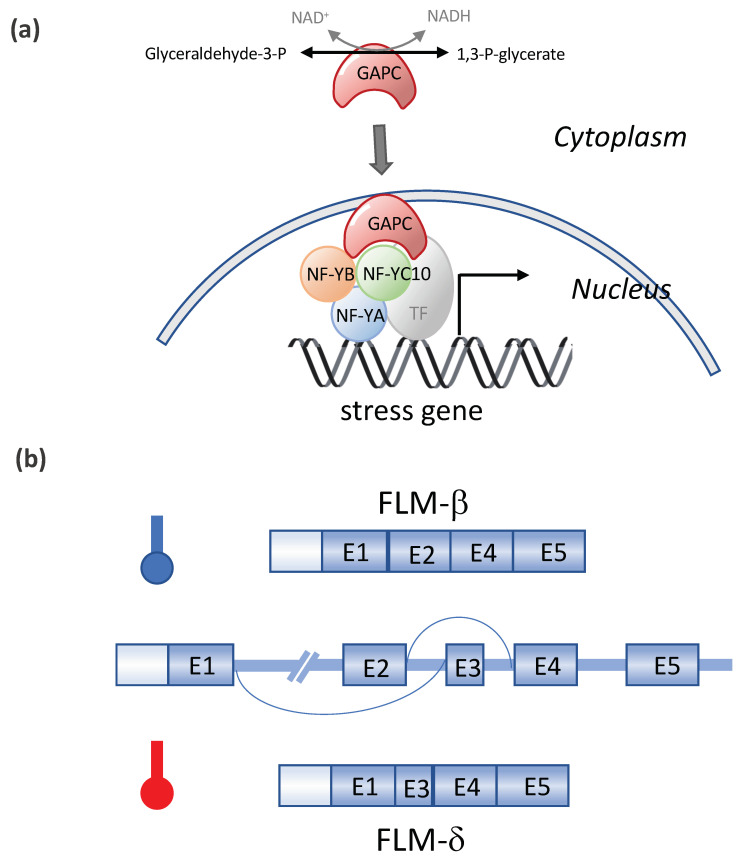
Moonlighting of cytosolic glyceraldehyde-3-phosphate dehydrogenase and the temperature-dependent alternative splicing of FLOWERING LOCUS M (FLM) in Arabidopsis. (**a**) Moonlighting of cytosolic glyceraldehyde-3-phosphate dehydrogenase (GAPC) (Based on Kim et al., 2020 [64]). The heat responses require GAPC and NF-YC10 interactions, and GAPC, which is normally located in the cytosol, enters the nucleus in response to heat. In the nucleus, GAPC binds to transcription factor NF-YC10 to increase the expression of heat-inducible genes, rendering Arabidopsis tolerant to heat stress. (**b**) Alternative splicing of FLM in response to the temperature. Modified from Capovilla et al., 2017 [74]. FLM has two splice variants, FLM-β and FLM-δ, in Arabidopsis, resulting from the temperature-dependent alternative splicing of FLM pre-mRNA via a mutually exclusive event (with Exon 2 or 3). The ratio of the two splicing isoforms, FLM-β/FLM-δ, fine-tunes the flowering time and is regulated by the ambient temperature.

**Table 1 ijms-22-00948-t001:** Signaling processes and pathways involved in thermotolerance.

Gene	Species	Main Results	References
HSR (heat stress response) -mediated transcriptional processes			
HSF1	Human	HSP (heat shock protein) 70 and the cochaperone Hdj1 interact directly with the transactivation domain of HSF (heat shock transcription factor) 1 and repress heat shock gene transcription	[23]
HSF1, HSF2	Human	Activation of HSF1 and HSF2 including the formation of trimers concomitant with the acquisition of DNA-binding activity and nuclear localization	[24]
HSF family	Multiple	Survey of Arabidopsis, rice and maize HSFs, sequence, structure and domain analysis	[27,28,29,30,31,32,33]
HSFA1s, HSFA2	Arabidopsis	*HSFA1d* and *HSFA1e* involved in the transcriptional regulation of *HSFA2* function as key regulators for the HSF signaling	[34]
HSFA2	Arabidopsis	HSFA2 function as key regulators of HSF signaling, *HSFA2* expression levels, heat stress responses and thermotolerance	[34,35,36,37,38,39]
HSFA6b	Arabidopsis	HSFA6b connects ABA (abscisic acid) signaling and ABA-mediated heat responses, contributes to thermotolerance, drought and salt tolerance	[40]
HSFBs	Arabidopsis	Heat-inducible expression of *HSFB1*, *HSFB2A*, and *HSFB2B* and the repressor activity of HSFBs	[41,42]
HSPs	Maize	HSPs response to heat stress in maize	[43,44,45,46,47,48]
UPR-mediated transcriptional processes			
bZIP28	Arabidopsis	bZIP (basic leucine zipper) 28 contributes to the upregulation of heat-responsive genes and heat tolerance	[49]
bZIP28, bZIP60	Arabidopsis	*bzip28 bzip60* double-mutant plants are sensitive to heat stress; bZIP28 binds directly to the promoters of heat-responsive genes	[50]
IRE1	Arabidopsis	Heat induces the splicing of bZIP60 by IRE1 (inositol-requiring enzyme 1); IRE1 confers heat stress tolerance, and this is attributable to its RNase domain	[51,52]
bZIP60, HSFTF13, HSPs	Maize	Maize heat stress response; bZIP60 links the UPR (unfolded protein response) to the HSR in maize by direct binding bZIP60 to the promoter of *Hsftf13* to active its expression	[14]
bZIP60	Maize	*bZIP60* expression levels in tropical and temperate maize lines with different heat tolerances	[53]
Others transcriptional processes			
ABA/ calcium/ SA/ethylene, etc.	Arabidopsis, maize	The involvement of calcium, ABA, ethylene, and SA (salicylic acid) in protecting against heat-induced oxidative damage in Arabidopsis or maize.	[54,55,56,57]
HSFA2c, HSPs, ABA	Tall Fescue, Arabidopsis	FaDREB (dehydration-responsive element-binding protein) 2, FaAREB3 directly bind to DRE/ABRE cis-elements in the *FaHSF2c* promoter to drive expression and improve heat tolerance in Tall Fescue and Arabidopsis	[58]
ERF1/JA/ ethylene	Arabidopsis	ERF (*ETHYLENE RESPONSE FACTOR*) 1 activates target genes by binding to GCC boxes or DRE/CRT elements in *HSFA3* and *HSP70*, and enhances tolerance to heat and other abiotic stresses.	[59]
JA/AOS/ ROS	Arabidopsis	JA (jasmonic acid) and ROS (reactive oxygen species) are required for plant acclimation to a combination of high light and heat stress, at both local and systemic level	[60,61]
DREB2A	Arabidopsis	Thermotolerance was significantly increased in plants overexpressing *DREB2A* CA and decreased in *DREB2A* knockout plants	[62]
DREB2A/ NF-YC10	Arabidopsis	As a DREB2A interactor, NF-YC10 (NUCLEAR FACTOR Y, SUBUNIT C 10) formed a transcriptional complex with NF-YA and NF-YB subunits, and the trimer enhanced heat stress-inducible gene expression in conjunction with DREB2A	[63]
NF-YC10/ GAPC	Arabidopsis	GAPC (glyceraldehyde-3-phosphate dehydrogenase) enters the nucleus in response to heat where it binds to NF- YC10 to increase the expression of heat-inducible genes, rendering Arabidopsis tolerant to heat stress.	[64]
SPL1, SPL12	Arabidopsis	SPL (SQUAMOSA PROMOTER BINDING PROTEIN-LIKE) 1 and SPL12 act redundantly to enhance thermotolerance through PYL-mediated ABA signaling in Arabidopsis	[65]
SDG25, ATX1	Arabidopsis	Histone H3K4 methyltransferases SDG25 and ATX1 regulate histone H3K4me3 level and prevent DNA methylation at loci associated with heat stress gene expression during stress recovery	[66]
APX1, 2	Arabidopsis	HSF21 is key in the early sensing of H_2_O_2_ stress in KO-*Apx1* (ascorbate peroxidase 1) plants. Increased H_2_O_2_ production during the early phase of heat stress is necessary for the HSFs-mediated induction of HSP17.6 and HSP18.2	[67,68,69]
Post-transcriptional processes			
AS, SR45a	Maize	Elevated maximum daily temperature induces alternative splicing and the roles of SR (serine/arginine-rich) 45a	[70]
HSF2A, HSFs	Arabidopsis, maize	HSF2A alternative splicing in response to heat stress	[30,71,72]
SR genes	Arabidopsis	Alternative splicing of SR genes in response to abiotic stress, including heat stress	[73]
FLM	Arabidopsis	Heat-induced alternative splicing of *FLM* (FLOWERING LOCUS M) isoforms and temperature dependent flowering	[74,75,76,77,78]
miR156-SPL	Arabidopsis	The miR156-SPL module mediates the response to recurring heat stress in Arabidopsis	[79]
miRNAs	Arabidopsis, maize	Identification of the miRNA in Arabidopsis and maize involved in post-transcriptional regulation of the response to heat stress	[80,81]

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
