# Peer review of "Heat Stress Responses and Thermotolerance in Maize"

_ijms, 2021, doi:10.3390/ijms22020948_

Round 1

Reviewer 1 Report

It is an interesting and well written article. During revisions authors should consider below mentioned suggestions:

1. There should be an introduction section after abstract. Perhaps authors forgot to mention introduction heading as there is some text already after abstract. Authors should elaborate bit more this introduction section.

2. Line 75, mention authors name in this reference for example Name et al. [12]

3. Subheading "2. HSR, a cytoplasmic heat stress response"  expand HSR here

4. Line 104; it would be better to specify gene names.

5. Figure 2 legend: provide full form for all abbreviation used.

6. same suggestion for figure 3

7. I suggest to elaborate a bit more section "3. Plant UPR in heat stress response "

8. It is a very good review and readers will expect a very good take home message from this article. So, I recommend authors to elaborate "Conclusion" section and focus more on future perspectives.

Author Response

Response to Reviewer 1 Comments

Comments and Suggestions for Authors

It is an interesting and well written article. During revisions authors should consider below mentioned suggestions:

  1. There should be an introduction section after abstract. Perhaps authors forgot to mention introduction heading as there is some text already after abstract. Authors should elaborate bit more this introduction section.

>We elaborated a bit more on the introduction (P1, lines 32-54) and the heading “Introduction” (P1, line 24) was added after the abstract, thanks.

P1, lines 32-54: “In recent years, comparative transcriptomics and metabolomics have been used to study heat stress responses in different maize lines. For example, in transcriptome studies of four heat-tolerant and four heat-susceptible inbred lines, 607 heat-responsive genes, as well as 39 heat-tolerance genes, were identified [3]. In another transcriptome study of heat tolerance in two sweet corn lines, the GO terms of biosynthesis of secondary metabolites, the upregulation of photosynthesis and the down regulation of ribosome function correlated with improved heat resistance [4]. In order to identify metabolic markers of heat stress responses, metabolite profiles of maize leaves have been analyzed under adverse environmental conditions including heat stress [5–7]. Similar to what has been found in Arabidopsis, altered energy pathways, increased production of branched-chain amino acids, raffinose family oligosaccharides, lipolysis products, and tocopherols have been found in response to heat stress [5–7]. Under severe stress conditions, the production has been observed of some metabolites that serve as osmolytes, antioxidants, and growth precursors and lipid metabolites that help protect membranes against heat stress.”

  1. Line 75, mention authors name in this reference for example Name et al. [12]

> We have made the change, thanks.

P3, line 109 “ for example Pooter et al., 2016 [17]” was used in the revised MS. The number of the line and ref were changed because the revised citation and content in the introduction section.

  1. Subheading "2. HSR, a cytoplasmic heat stress response"  expand HSR here

> We have made the change by using the full name in this subheading, thanks.

P4, line 127, “2. Heat stress response (HSR), a cytoplasmic heat stress response”

  1. Line 104; it would be better to specify gene names.

> Line 104 in the original MS is “and activate their transcription [20]. Plant HSF genes belong to moderate size families”, we are confused a with the comments. We specified “class A HSFs” instead of “HSFs” in line 146 (new, line 111 original MS).  If the comments were misunderstood, we are open to further revision.

  1. Figure 2 legend: provide full form for all abbreviation used.

>The full form of the abbreviation used were added in the figure legends as suggested, thanks.

P5, lines 168-170: “HSR, heat stress response; UPR, unfolded protein response; ER, endoplasmic reticulum; bZIP60, basic leucine zipper 60; HSP, heat shock protein; HSF, heat shock transcription factor; HSP70, heat shock protein 70; BIP, binding immunoglobulin protein; IRE1, inositol-requiring enzyme 1.” was added in the figure 2 legend.

  1. same suggestion for figure 3

The full form of the abbreviation used were added in the figure legends as suggested, thanks.

P9, lines 278-280: “HSFA, class A heat shock transcription factor; HSFB, class B heat shock transcription factor; ABA, abscisic acid; HSP, heat shock protein; HSF TF, heat shock transcription factor.” was added in the figure 3 legend.

  1. I suggest to elaborate a bit more section "3. Plant UPR in heat stress response "

>We elaborated a bit more of the section "3“, thanks.

P11, lines 356-360: “UPR is activated by misfolded proteins that accumulate in the ER under adverse environmental conditions. When the demand for protein folding exceeds the capacity of the system, UPR is initiated. Functional crosstalk between UPR and plant hormone signaling was also observed in plants to cope with the adverse environment [58,59].” was added in the section "3 Plant UPR in heat stress response”.

  1. It is a very good review and readers will expect a very good take home message from this article. So, I recommend authors to elaborate "Conclusion" section and focus more on future perspectives.

>We elaborated a bit more of the conclusion section and more on future perspectives.

P15, lines 544-552: A paragraph “Although our attention has been focused on individual genes in response to heat stress, we don’t have a holistic view of how heat tolerance in plants is brought about. Nonetheless, we are learning more about how plants sense temperature and turn it to a signal to control certain transcription, post-transcription, or translational events. It is of interest as to how two evolutionarily conserved cellular systems in the ER and cytoplasm communicate about heat stress with each other and elicit secondary signals involving calcium or ROS. We want to know how systemic signals are transmitted from leaves to “sink” organs such as the ear to regulate growth and development in response to adverse environment. Also, what are the differences in the way plants deal with heat stress or heat stress combined with drought, salinity, or other stresses?” was added to elaborate "Conclusion" section.

Reviewer 2 Report

The submitted manuscript “Heat stress responses and thermotolerance in maize” by authors Li Z., Howell S.H., represents a detailed and comprehensive review of available data concerning the heat responses of maize that can represent a valuable tool for breader to predict and identify heat tolerant lines in maize.

Development of plants and crops at higher than optimal temperature is one of the most widely spread environmental stress factors that disturb plant cellular homeostasis and decrease crop yield.

From the point of view of global warming summarizing the known information about the consequences of heat stress and plants’ response is of a great importance for better understanding the mechanisms of plants’ strategy to overcome and/or to acquire tolerance towards this environmental stress conditions.  

The response of maize to heat stress at a molecular level was discussed in respect to induction of stress-responsive genes and post-transcriptional processes.

The review is well structures, the figures and tables are informative. The manuscript is written in a good language.

I have only minor suggestions:

-Line 36-37 – the sentence is not clear

Line 38 – “decline 42%” to be changed to “decline with 42%”.

Line 177 – “suggest plants” to be changes to “suggest that plants”.

Line 256 – “JA was thought play” to be changed to “JA was thought to play”.

Line 307 – needs revision.

Author Response

Response to Reviewer 2 Comments

Comments and Suggestions for Authors

The submitted manuscript “Heat stress responses and thermotolerance in maize” by authors Li Z., Howell S.H., represents a detailed and comprehensive review of available data concerning the heat responses of maize that can represent a valuable tool for breeder to predict and identify heat tolerant lines in maize.

Development of plants and crops at higher than optimal temperature is one of the most widely spread environmental stress factors that disturb plant cellular homeostasis and decrease crop yield.

From the point of view of global warming summarizing the known information about the consequences of heat stress and plants’ response is of a great importance for better understanding the mechanisms of plants’ strategy to overcome and/or to acquire tolerance towards this environmental stress conditions.  

The response of maize to heat stress at a molecular level was discussed in respect to induction of stress-responsive genes and post-transcriptional processes.

The review is well structures, the figures and tables are informative. The manuscript is written in a good language.

I have only minor suggestions:

-Line 36-37 – the sentence is not clear

>We have made the change, thanks.

P2, lines 60-61 : “Lobell et al. [9] reported that for each degree day above 30 °C, the final yield of African maize declined 1%.” To “Lobell et al. [9] reported that for each degree day spent above 30 °C, the final yield of African maize declined 1%.”

Line 38 – “decline 42%” to be changed to “decline with 42%”.

>We have made the change, thanks. See P2, line 62.

Line 177 – “suggest plants” to be changes to “suggest that plants”.

>We have made the change, thanks. See P9, line 312.

Line 256 – “JA was thought play” to be changed to “JA was thought to play”.

>We have made the change, thanks. See P12, line 411.

Line 307 – needs revision.

>We have made the change, thanks.

“Other TFs involved in plant growth and development and other stress responses were also found to be involved in heat stress responses (Table 1).” was changed to “Other TFs and DNA methylation were also found to be involved in heat stress responses (Table 1).” In P12, line 418 in the revised MS.

Reviewer 3 Report

The present MS is very interesting, well-writen and demanding.

Although several studies have been published about heat stress induced changes in maize, these sutdies focused on morpho-physiological changes. A few of than, especially recent studies are about transcriptional changes, and a very few data are available about metabolite changes, in maize. However, in other corp plants there several recent studies based on microarray and metabolite analyses.

This review can show new direction for furhter investigation, so would encourage the authors to involve metabolite measurements, already in the introduction,too.

Some comments: 

  1. In section 1. line 34-44 Please describe in one sentence what are the most sensitive reasons of decline yield: pollen and pistil viability or embriogenesis. 
  2. Figure 1A: what do you mena on relative responses, related to what? what is indicated by the two blue narrows? Data represent the average of June and July? Figure 1B: please delete Zeitgeber from axis "x".
  3. The review is based on 107 references, but only 60 of them is from the last 10 years (2011-2021) Please consider more studies from the last decade, eg. Sun et al., Plant Biology, 2015; Obata et al. Plant Physiology, 2015; Essemine et al., Env. Exp. Bot, 2019; Shi et al., BMC PLant Biology, 2017; Abou-Deif et al., Bull Natl Res Cent , 2019. These studies are about metabolite profiles of maize under heat stress, metabolite and transcriptome changes, or proteomic analysis of HSPs in maize.
  4. The Table 1 is too large. I recommend to shorted it by focusing the most interesting and important data, for example the data on Drosophila can be omitted form the table, but can be further included in the text. Or mention only that rows and data, which are the most discussed in the text. As the most important relationship are presented in Figure 3A for Arabidopsis. Pegoraro et al., 2011 in J Crop Sci Biotech should be also involved in the references. 
  5. On Figure 3 B, the down and upregulation can be indcated by colours, in the present form it is not clear, what lighter and darker orange colour means in the table. Please also check the last sentence of the legend.
  6. In Section 2 line 152-179. Please more highlight the role of AtHSF6b in ABA signalling and ABA-mediated heat responses, in order to draw out hte attention for the possible importance of Hsftf13.
  7. Line 193, heart shock activites, you mean heat?
  8. In section 3. Please also see Poór et al., IJMS, 2019, about UPR.
  9. In Section 4, Please also see Casaretto et al., 2016 in BMC Genomics about the effects of expression of OsMYB55 in maize. 
  10. Conclusion, prospect and maize breeding should include more further perspective, e.g. about improvement of heat tolerance by for example lipid composition or by exogenous elicitors.
  11. It is characteristic of the whole manuscript that sentences that are too long and complicated make it difficult to follow.

    All these do not deduct from the value of the manuscript, which will fill a gap in the literature.

Author Response

Response to Reviewer 3 Comments

Comments and Suggestions for Authors

The present MS is very interesting, well-writen and demanding.

Although several studies have been published about heat stress induced changes in maize, these studies focused on morpho-physiological changes. A few of than, especially recent studies are about transcriptional changes, and a very few data are available about metabolite changes, in maize. However, in other crop plants there several recent studies based on microarray and metabolite analyses.

This review can show new direction for further investigation, so would encourage the authors to involve metabolite measurements, already in the introduction, too.

Some comments: 

  1. In section 1. line 34-44 Please describe in one sentence what are the most sensitive reasons of decline yield: pollen and pistil viability or embryogenesis. 

> We now state “The effects of high temperature during silking, pollination and grain filling have profound effects on maize pollen shed and viability [12,13] and at later stages on grain filling [10,14], P2, lines 64-68.

  1. Figure 1A: what do you mean on relative responses, related to what? what is indicated by the two blue narrows? Data represent the average of June and July? Figure 1B: please delete Zeitgeber from axis "x".

> It is called relative response, because the responses are relative to each other.  These are not absolute values; they are relative values.  For example, the response at 20°C is twice as great as that at 10°C.   The two blue narrows represent the average maximum daily temperature in Iowa during June and July.  Zeitgeber is important to include here because the chambers in which the experiments were conducted were offset from each other, so the times are virtual times.

  1. The review is based on 107 references, but only 60 of them is from the last 10 years (2011-2021) Please consider more studies from the last decade, eg. Sun et al., Plant Biology, 2015; Obata et al. Plant Physiology, 2015; Essemine et al., Env. Exp. Bot, 2019; Shi et al., BMC PLant Biology, 2017; Abou-Deif et al., Bull Natl Res Cent , 2019. These studies are about metabolite profiles of maize under heat stress, metabolite and transcriptome changes, or proteomic analysis of HSPs in maize.

>The references suggested by the reviewer and other two papers were added and discussed in the revised MS. Thanks for the suggestion.

For Frey et al., 2015; Shi et al., 2017; Obata et al., 2015; Sun et al., 2016; Essemine et al., 2020, they are cited in the introduction section: P1, lines 32-54 as references 3-7 in the revised MS. “In recent years, comparative transcriptomics and metabolomics have been used to study heat stress responses in different maize lines. For example, in transcriptome studies of four heat-tolerant and four heat-susceptible inbred lines, 607 heat-responsive genes, as well as 39 heat-tolerance genes, were identified [3]. In another transcriptome study of heat tolerance in two sweet corn lines, the GO terms of biosynthesis of secondary metabolites, the upregulation of photosynthesis and the down regulation of ribosome function correlated with improved heat resistance [4]. In order to identify metabolic markers of heat stress responses, metabolite profiles of maize leaves have been analyzed under adverse environmental conditions including heat stress [5–7]. Similar to what has been found in Arabidopsis, altered energy pathways, increased production of branched-chain amino acids, raffinose family oligosaccharides, lipolysis products, and tocopherols have been found in response to heat stress [5–7]. Under severe stress conditions, the production has been observed of some metabolites that serve as osmolytes, antioxidants, and growth precursors and lipid metabolites that help protect membranes against heat stress.”

For Abou-Deif et al., 2019 and Pegoraro et al., 2011, they are cited in P10, lines 332-334: “The proteins in maize plants produced in response to heat shock have been identified through proteome analysis and include such proteins as the ATPase beta subunit, HSP26, HSP16.9, and unknown HSP/chaperonin [54,55].” As references 54,55.

  1. The Table 1 is too large. I recommend to shorted it by focusing the most interesting and important data, for example the data on Drosophila can be omitted form the table, but can be further included in the text. Or mention only that rows and data, which are the most discussed in the text. As the most important relationship are presented in Figure 3A for Arabidopsis. Pegoraro et al., 2011 in J Crop Sci Biotech should be also involved in the references. 

>We have made the changes to the table. The data on Drosophila was deleted, and the rows with the similar results from different species were merged into one. See detail in table1.

  1. On Figure 3 B, the down and upregulation can be indicated by colours, in the present form it is not clear, what lighter and darker orange colour means in the table. Please also check the last sentence of the legend.

>We have made the change, thanks for the suggestion.

A revised Figure 3 was used. And the figure legend was revised to describe the colors indicated (P9, line 277 and line 278).

  1. In Section 2 line 152-179. Please more highlight the role of AtHSF6b in ABA signaling and ABA-mediated heat responses, in order to draw out hte attention for the possible importance of Hsftf13.

>We have made the changes, thanks for the suggestion.

P9, lines 286-289: “HSFA6b, responds to ABA signaling via the AREB/ABF-ABRE regulon, and activates transcriptional activity of HSP18.1-CI, DREB2A (dehydration-responsive element-binding protein 2A), and APX (cytosolic ascorbate peroxidase) 2 promoters [40].” Was inserted to highlight the role of AtHSF6b in ABA signaling.

  1. Line 193, heart shock activites, you mean heat?

>We have made the change from “heart” to “heat” (P10, line331), thanks for the suggestion.

  1. In section 3. Please also see Poór et al., IJMS, 2019, about UPR.

>We have added the citation as reference 58 and another Yang et al., 2017 as reference 59, thanks.

P11, lines 356-360: “UPR is activated by misfolded proteins that accumulate in the ER under adverse environmental conditions. When the demand for protein folding exceeds the capacity of the system, UPR is initiated. Functional crosstalk between UPR and plant hormone signaling was also observed in plants to cope with the adverse environment [58,59].” was added in the section "3 Plant UPR in heat stress response”.

  1. In Section 4, Please also see Casaretto et al., 2016 in BMC Genomics about the effects of expression of OsMYB55 in maize. 

>We have added the citation as reference 79, thanks.

P12, lines 436-439: “Over-expression of OsMYB55 in maize improves plant tolerance to high temperature and water stress during vegetative growth by activation of stress-responsive genes such as thaumatin, Hsftf19, two WRKY proteins, and five MYB proteins genes [79].”

  1. Conclusion, prospect and maize breeding should include more further perspective, e.g. about improvement of heat tolerance by for example lipid composition or by exogenous elicitors.

>Thank you for the suggestions.  We have elaborated a bit more of the conclusion" section.  However, we have not included discussions about lipid composition and exogenous elicitiors, because we have tried to stay within the context of our review, which did not cover these areas.

P15, lines 544-552: A paragraph “Although our attention has been focused on individual genes in response to heat stress, we don’t have a holistic view of how heat tolerance in plants is brought about. Nonetheless, we are learning more about how plants sense temperature and turn it to a signal to control certain transcription, post-transcription, or translational events. It is of interest as to how two evolutionarily conserved cellular systems in the ER and cytoplasm communicate about heat stress with each other and elicit secondary signals involving calcium or ROS. We want to know how systemic signals are transmitted from leaves to “sink” organs such as the ear to regulate growth and development in response to adverse environment. Also, what are the differences in the way plants deal with heat stress or heat stress combined with drought, salinity, or other stresses?” was added to elaborate "Conclusion" section.

  1. It is characteristic of the whole manuscript that sentences that are too long and complicated make it difficult to follow.

>We fixed some of them as indicated by the "Track Changes" function. We hope that the variation in sentence length throughout the review helps the reader in reading the article.  If the editors wish to make some stylisitic suggestion, we are open to further revision.

All these do not deduct from the value of the manuscript, which will fill a gap in the literature.

Round 2

Reviewer 1 Report

I have no further comments on the revised version.

Reviewer 3 Report

The authors have thoroughly rewritten the MS as requested.